# Torsionally-Induced Stability in Spinors

Luca Fabbri

DIME, Sez. Metodi e Modelli Matematici, Università di Genova, Via all'Opera Pia 15, 16145 Genova, Italy; fabbri@dime.unige.it

**Abstract:** We consider the role of the torsion axial-vector played in the dynamics of the Dirac spinor fields: we show that the torsional correction entails effects that render regular the otherwise singular spinorial matter field distribution. Comments about consequences for physics are eventually given.

**Keywords:** torsion axial-vector; Dirac spinor fields; singularity problem

## 1. Introduction

Einstein's theory of gravitation, or more in general any of its extensions with higher-order derivative terms, have acquired over the years the status of superb physical theories: established upon one single principle, that is that the space-time is a $(1+3)$-dimensional manifold, whose curvature is determined by the energy content, they have produced many of the most surprising predictions in all of physics, starting with the bending of light rays in 1916 and ending with the detection of gravitational waves one century later. In fact, if Dark Matter is indeed a form of matter, there is not a single prediction left to confirm.[1]

Quantum Field Theory has done no less: its predictions are among the most precise in history, and while still in need of rigorous mathematical formalization very few doubt that this task will be accomplished in the foreseeable future.

There are, nevertheless, still two problems that appear to cloud the beauty of these two pillars of modern physics, both related to the issue of singularities. QFT's range of applicability, while very vast, is limited to the assumption that particles be structureless point-like objects. Einstein gravity, on the other hand, seems to be doomed by what is known as the Hawking–Penrose theorem, stating that gravitational formation of singularities of matter distributions is an unavoidable consequence for all known fields.

In detail, the HP theorem tells that if some conditions on the energy density tensor hold, as they do for all physical fields, then the overall gravitational pull would never stop and the whole material distribution would inevitably collapse into a singular point. Since the HP theorem, in its essence, is a result of Einstein gravity, a first attempt at avoiding singularities might come for extensions of Einstein gravity, where higher-order derivative field equations would entirely change the shape of the problem. It is indeed the case that at least in some of these extensions there are solutions of the gravitational field equations that are without singularities [1]. Because the theory presented in [1] is the only extension of gravity that is renormalizable [2], one may be led to think that all singularity issues are avoidable also in QFT and therefore that nothing more need be done. Just the same, even if this were to be true, one would still have to rely upon the theory presented in [1]. Granted that such an extension of Einstein gravity is the only one compatible with renormalizability, we still do not know if such a theory is the description of gravity at high energy densities.

In lack of any prediction for a high-energy density gravitational effect, one might wish to stay on the more comfortable environment provided by Einstein gravity. However, it in, the problem of singularity formation would be still far from being solved. So a possible, alternative way out of the problem might be to look for different manners in which Einstein gravity could be enlarged. As Einstein gravity is indeed a remarkably stringent theory, there are not many possibilities. In fact, in the perspective of staying on $(1+3)$-dimensional

manifolds, and without tampering with the differential order of the field equations, the only possibility that remains is not to neglect the torsion.

The torsion of space-time is a very natural ingredient of differential geometry, one that is always present in the most general setting. For a comprehensive review on torsion we invite the interested reader to have a look at the introductory chapter on the present Special Issue [3] and at some of the references therein, either for a general review [4] or for the seminal papers [5,6]. As the torsional completion of Einstein gravity has the same differential order of Einstein gravity itself then the HP theorem applies as usual and the presence of torsion can only change the conditions on the energy density. This was the initial hope, as it was worked out by Kerlick in [7]. The initial hope, however, was soon to be lost, since Kerlick himself pointed out that torsion, instead of solving this problem, was actually worsening it [7,8]. The catch was that the model used was just the torsional completion of Einstein gravitation known as Einstein–Sciama–Kibble theory, in which the torsion–spin interaction is repulsive [9]. While such a repulsive character may at first look appealing, in view of avoiding singularities, in reality repulsive forces give positive energy contributions that increase the gravitational pull and so the tendency to form these singularities. This is the physical reason why Kerlick and Inomata find that singularities are enhanced in [7,8]. In [9] Popławski points out that, despite this, singularities still do not appear at least in some cosmological scenarios.

It is difficult to understand why this should be the case, and instead of a theory in which singularities are avoided despite being enhanced, it would be better to have a theory where singularities are avoided because they are dissipated. To this purpose, one must then violate all of the energy conditions. This can only be done by lowering the energy contributions via the inclusion of negative potentials, and these can only be given by some attractive force. Thus, as paradoxical as it may be, one has to look for attractive torsionally-induced spin–spin interaction in the Dirac equation. This is precisely what many have done, for example in [10–12] and references therein.

The idea is that the ESK theory, of all torsional completions of Einstein gravity, is only the simplest, the one in which torsion is added only implicitly through the curvature tensor in the Lagrangian. That is, from Einstein gravity $L = R(g)$ the ESK theory is given by $L = R(g, Q)$ with $Q$ being the torsion tensor. In [10] the authors take instead $L = R(g, Q) + Q^2$ where $Q^2$ is formally any of the three scalar contractions of the squared torsion tensor. Thus, with the freedom granted by three new universal constants, the authors of [10] are able to obtain an effective constant for torsion whose sign is undetermined and therefore with the possibility of being inverted with respect to the ESK theory in its simplest form, and as a result they indeed get to avoid the singularity of the early universe.

Others, however, took into account yet another generalization. Because the $Q^2$ term can be read as a combination of three mass terms, one for each of the three irreducible components of torsion, we could then make sure that its sign be that of a positive mass, then add the three dynamical terms of the three parts of torsion. In reality, however, of the three irreducible components of torsion, only the completely antisymmetric part seems to be acceptable for consistency arguments connected to the metric-compatibility of the manifold and related properties [13–16]. Because a torsion that is completely antisymmetric is the Hodge dual of an axial-vector, the resulting model is that of an axial-vector torsion, namely a model in which torsion is a parity-odd Proca field. In it there may be effects that are due to torsion propagation [17]. Some of these have been studied for the avoidance of singularity [18].

In the present work, we will recall all results of [18] but employing the polar form of spinors [19] so to get cleaner and more powerful results for the problem of singularities in gravitational systems. In addition, we will extend all of the above results to the case of singularities that appear also in absence of gravity and for general systems.

## 2. The Polar Form of Spinors

To do what we intend to do we first introduce the polar formulation of Dirac spinors, for which we shall always refer to the fundamental work [19] and references therein.

However, we recall here a few basic conventions. The set of Clifford matrices $\gamma_a$ verifying $\{\gamma_a, \gamma_b\} = 2\eta_{ab}\mathbb{I}$ are used to set $\sigma_{ab} = \frac{1}{4}[\gamma_a, \gamma_b]$ as the generators of the complex Lorentz transformation. Then $2i\sigma_{ab} = \varepsilon_{abcd}\boldsymbol{\pi}\sigma^{cd}$ is used to define the matrix $\boldsymbol{\pi}$ so that $\mathbb{I}, \gamma_a$, $\sigma_{ab}, \gamma_a\boldsymbol{\pi}, \boldsymbol{\pi}$ are a basis for the space of $4 \times 4$ complex matrices. As a consequence, we have that the identities $\gamma_i\gamma_j\gamma_k = \gamma_i\eta_{jk} - \gamma_j\eta_{ik} + \gamma_k\eta_{ij} + i\varepsilon_{ijkq}\boldsymbol{\pi}\gamma^q$ hold in general.

The product of a complex Lorentz transformation and a unitary phase is the spin transformation $S$ and spinors $\psi$ are defined as what transforms according to these spin transformations. Adjoint spinors are related by $\overline{\psi} = \psi^\dagger\gamma^0$ and with these one can construct all spinor bi-linears and prove their Fierz identities. Before doing that however we will give the following theorem. If $|\overline{\psi}\psi|^2 + |i\overline{\psi}\boldsymbol{\pi}\psi|^2 \neq 0$ one can always write the spinor field as

$$\psi = \phi e^{-\frac{i}{2}\beta\boldsymbol{\pi}} L^{-1} \begin{pmatrix} 1 \\ 0 \\ 1 \\ 0 \end{pmatrix} \tag{1}$$

up to $\psi \to \boldsymbol{\pi}\psi$ and for some $L$ having the structure of a spin transformation whose parameters are recognized to be the spinorial Goldstone fields and where $\phi$ and $\beta$ are a real scalar and a real pseudo-scalar field called module and chiral angle. The above is the polar form of the spinor field and with it we can write the spinor bi-linears as

$$i\overline{\psi}\boldsymbol{\pi}\psi = 2\phi^2 \sin\beta \tag{2}$$
$$\overline{\psi}\psi = 2\phi^2 \cos\beta \tag{3}$$

as well as

$$\overline{\psi}\gamma^a\boldsymbol{\pi}\psi = 2\phi^2 s^a \tag{4}$$
$$\overline{\psi}\gamma^a\psi = 2\phi^2 u^a \tag{5}$$

known as spin axial-vector and velocity vector verifying

$$u_{[\mu}s_{\nu]}\sigma^{\mu\nu}\boldsymbol{\pi}\psi + \psi = 0 \tag{6}$$
$$u_\mu\gamma^\mu\psi = -s_\mu\gamma^\mu\boldsymbol{\pi}\psi = e^{i\beta\boldsymbol{\pi}}\psi \tag{7}$$

and

$$u_a u^a = -s_a s^a = 1 \tag{8}$$
$$u_a s^a = 0 \tag{9}$$

known as Fierz identities. Consequently, when the spinor field is re-written in polar form its eight real components are re-arranged so that the two degrees of freedom, the module and chiral angle, remain isolated from the six components that can always be transferred into the frame, and that is what is known to be the Goldstone fields.

To better see the role of these Goldstone fields, we can write them explicitly according to the formula

$$L^{-1}\partial_\mu L = iq\partial_\mu\xi\mathbb{I} + \frac{1}{2}\partial_\mu\xi^{ab}\sigma_{ab} \tag{10}$$

in which $\xi$ and $\xi^{ab}$ are respectively the gauge and space-time Goldstone fields. Together with the gauge potential and spin connection $A_\mu$ and $C_{ij\mu}$ we can define

$$q(\partial_\mu \xi - A_\mu) \equiv P_\mu \tag{11}$$
$$\partial_\mu \xi_{ij} - C_{ij\mu} \equiv R_{ij\mu} \tag{12}$$

which can be proven to be real tensors respectively called gauge and space-time tensorial connection. With them

$$\boldsymbol{\nabla}_\mu \psi = \left( \nabla_\mu \ln \phi \mathbb{I} - \tfrac{i}{2} \nabla_\mu \beta \boldsymbol{\pi} - i P_\mu \mathbb{I} - \tfrac{1}{2} R_{ij\mu} \sigma^{ij} \right) \psi \tag{13}$$

is the polar form of the covariant derivative of the spinor field. As a consequence

$$\nabla_\mu s_i = R_{ji\mu} s^j \tag{14}$$
$$\nabla_\mu u_i = R_{ji\mu} u^j \tag{15}$$

as general identities. As is clear, after that the Goldstone fields are transferred into the phase and the frame, they combine with gauge potential and spin connection hence becoming the longitudinal parts of the $P_\mu$ and $R_{ij\mu}$ tensors.

To write the Dirac spinor field equations for the polar variables we introduce the pair of dual potentials

$$\Sigma_{ij\mu} = R_{ij\mu} - 2P_\mu u^a s^b \varepsilon_{ijab} \tag{16}$$
$$M^{ab}{}_\mu = \tfrac{1}{2} R_{ij\mu} \varepsilon^{ijab} + 2P_\mu u^{[a} s^{b]} \tag{17}$$

and their contractions $\Sigma_{\alpha\nu\pi} g^{\nu\pi} = \Sigma_\alpha$ and $M_{\alpha\nu\pi} g^{\nu\pi} = M_\alpha$ in terms of which we have

$$\nabla_\mu \ln \phi^2 + \Sigma_\mu + 2m s_\mu \sin \beta = 0 \tag{18}$$
$$\nabla_\mu \beta - 2X W_\mu + M_\mu + 2m s_\mu \cos \beta = 0 \tag{19}$$

with $m$ being the mass of the spinor, $X$ the spin–torsion coupling constant and $W_\mu$ the torsion axial-vector [3,13–16].

These matter field equations have to be complemented with the geometric field equations given by

$$\nabla_\rho (\partial W)^{\rho\mu} + M^2 W^\mu = 2X \phi^2 s^\mu \tag{20}$$

where $M$ is the mass of torsion together with

$$\begin{aligned}
R^{\rho\sigma} - \tfrac{1}{2} R g^{\rho\sigma} - \Lambda g^{\rho\sigma} = \tfrac{1}{2} \Big[ &\tfrac{1}{4} F^2 g^{\rho\sigma} - F^{\rho\alpha} F^\sigma{}_\alpha + \\
&+ \tfrac{1}{4} (\partial W)^2 g^{\rho\sigma} - (\partial W)^{\sigma\alpha} (\partial W)^\rho{}_\alpha + \\
&+ M^2 (W^\rho W^\sigma - \tfrac{1}{2} W^2 g^{\rho\sigma}) + \\
&+ \phi^2 [P^\rho u^\sigma + P^\sigma u^\rho + \\
&+ (\nabla^\rho \beta / 2 - X W^\rho) s^\sigma + (\nabla^\sigma \beta / 2 - X W^\sigma) s^\rho - \\
&- \tfrac{1}{4} R_{\alpha\nu}{}^\sigma s_\kappa \varepsilon^{\rho\alpha\nu\kappa} - \tfrac{1}{4} R_{\alpha\nu}{}^\rho s_\kappa \varepsilon^{\sigma\alpha\nu\kappa} ] \Big]
\end{aligned} \tag{21}$$

for the gravitational field equations and

$$\nabla_\sigma F^{\sigma\mu} = 2q \phi^2 u^\mu \tag{22}$$

for the electrodynamic field equations.

As a concluding remark we also give the expression

$$L = -\tfrac{1}{4}(\partial W)^2 + \tfrac{1}{2}M^2W^2 - R - 2\Lambda - \tfrac{1}{4}F^2$$
$$+ \tfrac{i}{2}(\overline{\psi}\gamma^\mu\boldsymbol{\nabla}_\mu\psi - \boldsymbol{\nabla}_\mu\overline{\psi}\gamma^\mu\psi) -$$
$$- X W_\sigma \overline{\psi}\gamma^\sigma\boldsymbol{\pi}\psi - m\overline{\psi}\psi \tag{23}$$

which is the Lagrangian of the full theory.

The field equations above are just the polar form of the field equations one would have in the torsional completion of gravity with electrodynamics [3].

We have maintained the Lagrangian in its standard form to easily compare it to the Lagrangians of known theories, as we will do next.

### 3. Singularities Avoidance in Spinorial Average

We begin a more in-depth treatment of the theory, useful for future developments, by introducing the concept of effective approximation. As we have just seen, torsion can be massive, and since we have not observed it so far, it may also be argued that its mass should be quite large indeed. So it is reasonable to assume that there might be regimes in which the mass term dominates all dynamical terms. When the dynamical terms can be suppressed in favour of the mass term we are in the so-called effective approximation. The torsion field equations reduce to

$$M^2 W^\mu \approx 2X\phi^2 s^\mu \tag{24}$$

and so by substituting torsion everywhere in terms of the spin we should expect torsion–spin interactions to convert into spin–spin interactions of non-linear character.

When this is done in the Lagrangian we get

$$L = -R - 2\Lambda - \tfrac{1}{4}F^2$$
$$+ \tfrac{i}{2}(\overline{\psi}\gamma^\mu\boldsymbol{\nabla}_\mu\psi - \boldsymbol{\nabla}_\mu\overline{\psi}\gamma^\mu\psi) -$$
$$- \tfrac{1}{2}\tfrac{X^2}{M^2}\overline{\psi}\gamma^\sigma\boldsymbol{\pi}\psi\overline{\psi}\gamma_\sigma\boldsymbol{\pi}\psi - m\overline{\psi}\psi \tag{25}$$

or by employing Fierz re-arrangements

$$L = -R - 2\Lambda - \tfrac{1}{4}F^2$$
$$+ \tfrac{i}{2}(\overline{\psi}\gamma^\mu\boldsymbol{\nabla}_\mu\psi - \boldsymbol{\nabla}_\mu\overline{\psi}\gamma^\mu\psi) +$$
$$+ \tfrac{1}{2}\tfrac{X^2}{M^2}\overline{\psi}\gamma^\sigma\psi\overline{\psi}\gamma_\sigma\psi - m\overline{\psi}\psi \tag{26}$$

or yet again

$$L = -R - 2\Lambda - \tfrac{1}{4}F^2$$
$$+ \tfrac{i}{2}(\overline{\psi}\gamma^\mu\boldsymbol{\nabla}_\mu\psi - \boldsymbol{\nabla}_\mu\overline{\psi}\gamma^\mu\psi) +$$
$$+ \tfrac{1}{2}\tfrac{X^2}{M^2}|i\overline{\psi}\boldsymbol{\pi}\psi|^2 + \tfrac{1}{2}\tfrac{X^2}{M^2}|\overline{\psi}\psi|^2 - m\overline{\psi}\psi \tag{27}$$

revealing that torsionally-induced spin–spin interactions are attractive. In fact if we were to further split into the left-handed and right-handed projections then we would see that such an attraction takes place between these two chiral parts. This is expected, since the above Lagrangian is just the Lagrangian of the Nambu–Jona-Lasinio model [20].

The effective approximation within the gravitational field Equation (21) gives instead the expression

$$R^{\rho\sigma} - \tfrac{1}{2}Rg^{\rho\sigma} - \Lambda g^{\rho\sigma} = \tfrac{1}{2}\big[\tfrac{1}{4}F^2 g^{\rho\sigma} - F^{\rho\alpha}F^\sigma{}_\alpha +$$
$$+ \phi^2(P^\rho u^\sigma + P^\sigma u^\rho + \nabla^\rho\beta/2s^\sigma + \nabla^\sigma\beta/2s^\rho -$$
$$- \tfrac{1}{4}R_{\alpha\nu}{}^\sigma s_\kappa \varepsilon^{\rho\alpha\nu\kappa} - \tfrac{1}{4}R_{\alpha\nu}{}^\rho s_\kappa \varepsilon^{\sigma\alpha\nu\kappa} + 2\tfrac{X^2}{M^2}\phi^2 g^{\rho\sigma})\big] \tag{28}$$

while in the Dirac Equation (19) we get

$$\nabla_\mu \beta - 2(2\tfrac{X^2}{M^2}\phi^2 - m\cos\beta)s_\mu + M_\mu = 0 \tag{29}$$

with all other equations remaining unchanged. Contracting (28) and employing the scalar product of (29) along the spin we get the particularly simple

$$R + 4\Lambda = -2\tfrac{X^2}{M^2}\phi^4 - m\phi^2\cos\beta \tag{30}$$

which can be substituted back into the initial field equations to give the gravitational field equations

$$
\begin{aligned}
R^{\rho\sigma} = & -\Lambda g^{\rho\sigma} + \tfrac{1}{2}[\tfrac{1}{4}F^2 g^{\rho\sigma} - F^{\rho\alpha}F^\sigma{}_\alpha + \\
& + \phi^2(P^\rho u^\sigma + P^\sigma u^\rho + \nabla^\rho\beta/2s^\sigma + \nabla^\sigma\beta/2s^\rho - \\
& - \tfrac{1}{4}R_{\alpha\nu}{}^\sigma s_\kappa \varepsilon^{\rho\alpha\nu\kappa} - \tfrac{1}{4}R_{\alpha\nu}{}^\rho s_\kappa \varepsilon^{\sigma\alpha\nu\kappa} - m\cos\beta g^{\rho\sigma})]
\end{aligned}
\tag{31}
$$

so that all spin–spin interactions disappear as source for the dynamics of the Ricci tensor.

This however does not mean that torsion has no impact, as we will now see.

The gravitational field equations written for the Ricci tensor are the most adapted to investigate the dominant energy condition since this is given as

$$R^{\rho\sigma}u_\rho u_\sigma \geqslant 0 \tag{32}$$

with $u^\alpha$ a normalized time-like vector. Imposing this condition on (31) we obtain

$$
\begin{aligned}
\phi^2(-2\tfrac{X^2}{M^2}\phi^2 - s^\mu\nabla_\mu\beta/2 + \tfrac{1}{4}R_{ijb}s_a\varepsilon^{ijba} - \\
- \tfrac{1}{4}R_{\alpha\nu\sigma}u^\sigma u_\rho s_\kappa \varepsilon^{\alpha\nu\rho\kappa} + \tfrac{1}{2}m\cos\beta) - \Lambda \geqslant 0
\end{aligned}
\tag{33}
$$

where (29) was again used and in which electrodynamics has been neglected for simplicity.

This expression is still quite general for the purpose that we have in mind.

In the treatment done by Kerlick and further generalizations [7–12], this is the moment where some additional assumptions are made by these authors. They are either specific symmetries or special properties for the material distributions. Because most of these assumptions appear ad hoc and some of them even unphysical, we will assume none of them in the following. In what is next we employ the only assumption that is physically adequate and very general, namely for gravitating systems such as the Big Bang and Black Holes the random distribution of particles allows the spin average to vanish all terms in which spin appears linearly. This means that (33) becomes

$$-2\tfrac{X^2}{M^2}\phi^4 + \tfrac{1}{2}m\phi^2\cos\beta - \Lambda \geqslant 0 \tag{34}$$

in spin-average. When no matter field is present we have that (34) reduces to $-\Lambda \geqslant 0$ which is obviously invalid if the cosmological constant is positive (as is in our convention). This is what we expect for a universe in continuous accelerated expansion. Neglecting the cosmological constant

$$-4\tfrac{X^2}{M^2}\phi^2 + m\cos\beta \geqslant 0 \tag{35}$$

as easy to see. In absence of torsion this is always verified whenever $m\cos\beta \geqslant 0$ as is always the case for a standard matter distribution (for which the chiral angle is assumed to vanish albeit not in a justified way [19]). This is what recovers the common arguments about the gravitational formation of singularities and their inevitability. In presence of torsion,

however, we have that for densities that are larger and larger, regardless of the values of the chiral angle, we can always approximate

$$-\frac{X^2}{M^2} \geqslant 0 \tag{36}$$

again as is clear. This is of course always violated in a very dramatic way (see for instance [18]). So failing the hypothesis there is no implication from the HP theorem.

The conclusion is that for systems given by statistically distributed spinor fields, such as the Big Bang or Black Holes, the formation of gravitational singularities is not a necessity.

It is instructive to look back at the previous attempts in a comparative way and draw the differences. That is, it is now the time to ask, given that the torsionally-induced spin–spin interaction renders the gravitational formation of singularities not a necessary occurrence, how is it that in the past previous attempts did not come up with the same result? The answer is that there are many different ways to let torsion take its place in the Lagrangian: one, the simplest, is via the minimal coupling, as done by the Kerlick school [7,9]; another is via the generalization that takes into account also explicit squared torsion terms in the Lagrangian, as in [10]; the final is to include also all squared derivative torsion terms within the most general dynamically-consistent Lagrangian [18]. In the first one, which is just the ESK gravitation, the torsionally-induced spin–spin interaction has a constant equal to $3/16$ when measured in natural units, since its value is rigidly linked to that of the Newton constant; in the generalizations of the ESK gravity the constant has an indefinite value, as it results from combining the three constants that appear in front of each of the three squared torsion terms that may be added; in the most general case the constant is $-X^2/M^2$ when the effective approximation is implemented because in a theory in which torsion propagates we must require the energy density and the mass to be positive. Therefore the gravitational formation of singularities that cannot be avoided in Einstein gravity, and that is worsened in ESK gravity, may become avoidable in further generalizations, and it is certainly avoidable in the most general case of propagating torsion (in effective approximation).

The mechanism for gravitational singularity avoidance can only be effectively enforced in a theory of propagating torsion, that is in the most general physical situation.

The interpretation we can give is that in such a case the torsionally-induced spin–spin interactions turn attractive, which have a negative potential, and thus the overall energy contribution is decreased. If the negative potential becomes dominant, the total energy turns negative, then the curvature reverts sign, and gravity becomes repulsive.

It is the fact that torsion is an attraction between chiral parts what causes the relaxation of the attraction of the gravitational field of the overall material distribution.

## 4. Singularities Avoidance for Single Particles

In the previous section we have investigated the problem of the singularity formation in gravitational systems, constituted by statistically distributed spinor fields. The vanishing of the spin-average in linear terms was our only hypothesis.

What if we have no random distributions? Worse, what if we have a single particle as in QFT? In this case gravitation would always be negligible and we could not use the above argument to avoid singularities, so what is there to be done in this case?

Of course, the full analysis has to be re-done without the help of any of the previous hypotheses, and when only torsion appears in the dynamics of spinor fields. As a consequence, in the following we will consider only torsion and Dirac spinor field equations and see what information we can extract for the singularity problem. Torsion will still be taken in its effective approximation and for the Dirac spinor field equations we will do some formal manipulations to put them in a form that is more suitable for our purposes.

One of the advantages of the polar formulation of the Dirac theory is that, with it, it is very easy to see that the second-order derivative field equations are given by

$$\nabla^\mu(\phi^2\nabla_\mu\beta)-(8X^2/M^2\phi^2 m\sin\beta-$$
$$-2XW^\mu\Sigma_\mu-\nabla_\mu M^\mu+M_\mu\Sigma^\mu)\phi^2=0 \tag{37}$$

as a continuity equation for the chiral angle and

$$|\nabla\beta/2|^2-m^2-\phi^{-1}\nabla^2\phi+\tfrac{1}{4}(-2\nabla_\mu\Sigma^\mu+$$
$$+\Sigma^\mu\Sigma_\mu-M_\mu M^\mu+4XM_\mu W^\mu-4X^2W^\mu W_\mu)=0 \tag{38}$$

as a Hamilton–Jacobi equation for the module. Singularities form at high density, and so what really interests us is the Hamilton–Jacobi equation for the module (38), for which we are now going to assume torsion in its effective approximation, resulting into the simpler expression

$$\nabla^2\phi-4\frac{X^4}{M^4}\phi^5-2\frac{X^2}{M^2}(M_\mu s^\mu)\phi^3+(\tfrac{1}{2}\nabla_\mu\Sigma^\mu-$$
$$-\tfrac{1}{4}\Sigma^\mu\Sigma_\mu+\tfrac{1}{4}M_\mu M^\mu-|\nabla\beta/2|^2+m^2)\phi=0 \tag{39}$$

in which as expected non-linearities have arisen. In it, the quintic term is always negative, so that the highest-order self-interaction is always attractive. Then the cubic term is negative so long as $M_\mu s^\mu>0$ and in this case also the lower-order self-interaction is attractive. The linear term is positive when $2\nabla_\mu\Sigma^\mu-\Sigma^\mu\Sigma_\mu+M_\mu M^\mu-|\nabla\beta|^2+4m^2>0$ and in this case it behaves as a regular mass term. In case of singularity formation, we should expect the density to increase, leading to larger values of the module. There is no need to evaluate the signs of these two terms because both are negligible with respect to the highest-order term and therefore we end up having the definite form

$$\nabla^2\phi-4\frac{X^4}{M^4}\phi^5=0 \tag{40}$$

which does have a non-singular behaviour. In fact it is a straightforward operation to see that in stationary cases

$$\vec{\nabla}\cdot\vec{\nabla}\phi+4\frac{X^4}{M^4}\phi^5=0 \tag{41}$$

and for spherical symmetry

$$\frac{1}{r^2}\partial_r(r^2\partial_r\phi)+4\frac{X^4}{M^4}\phi^5=0 \tag{42}$$

which is exactly one of the instances of the Lane–Emden equation that can be solved. The explicit solution is

$$\phi=\sqrt{\frac{3M^4}{4X^4+3M^4r^2}} \tag{43}$$

which is non-singular. Notice that when the torsion constant tends to zero then the $1/r$ singular behaviour is recovered as a general feature of the solution. Therefore we can say that it is precisely the presence of torsion what forbids singularities.

The formation of singularities recovered with a vanishing torsion constant can also be seen as due to a too large torsion mass. It is therefore tempting to speculate that it is the torsion mass what gives the scale of the cut-off beyond which divergences are negligible in QFT. In more detail, renormalization schemes are based on the idea for which ultraviolet divergences appear because we do not actually know what happens at very high energy, but we postulate that in these regimes physics is different and free of singularities. As it

stands one may think that this new physics is just what we would have always had, had we never neglected torsion from the beginning.

How would QFT change if we were to employ the non-singular solutions we have in presence of torsion? Would we still need any of the known renormalization protocols?

We conclude this section by noticing that if in Equation (39) both conditions on the cubic and linear terms hold then (39) has the structure of a solitonic equation.

### 5. One Specific Circumstance: Walking Droplets

Let us focus more on (39) and ask the following question: what happens if in it the linear term is $2\nabla_\mu \Sigma^\mu - \Sigma^\mu \Sigma_\mu + M_\mu M^\mu - |\nabla \beta|^2 + 4m^2 < 0$ instead? For such a situation the linear term becomes that of a mass with imaginary value. At high density solutions are still regular. However far from the origin there no longer is an exponentially decreasing behaviour with radial distance but rather an oscillating behaviour. Solutions would no longer be solitons but something more similar to a material distribution with a solid core surrounded by wavelets. It is tempting to see in these the bouncing droplets used for hydrodynamic analogs of quantum systems [21,22].

Let us try to give an explicit example. Consider $P_t = m$ and a tensorial connection with component

$$R_{r\varphi\varphi} = -r|\sin\theta|^2 \tag{44}$$
$$R_{r\theta\theta} = -r \tag{45}$$
$$R_{\varphi\theta t} = -2\varepsilon r^2 \sin\theta \tag{46}$$

where $\varepsilon$ is a constant due to the Riemann curvature being equal to zreo. These are compatible with $s_r = 1$ and $u_t = 1$ from the constraints (14) and (15). The dual potentials (16) and (17) are given by the expressions

$$\Sigma_r = 2/r \tag{47}$$

$$M_r = -2(m+\varepsilon) \tag{48}$$

from which we can now see that for $\beta = 0$ they give

$$s^\mu M_\mu = 2(m+\varepsilon) \tag{49}$$
$$2\nabla_\mu \Sigma^\mu - \Sigma^\mu \Sigma_\mu + M^\mu M_\mu + 4m^2 = -4(2m+\varepsilon)\varepsilon \tag{50}$$

to be discussed. If $\varepsilon$ is negative with $m > |\varepsilon| > 0$ we obtain the self-interaction to be attractive and the mass term to be a real mass term, so that the conditions to have solitonic solutions are ensured. If instead $\varepsilon$ is positive we always obtain a self-interaction of attractive character but now with mass of imaginary type, so that the solution has the peripheral oscillations proper of wavelets. Equation (38) in this case is in fact given according to

$$\nabla^2 \phi - 4\frac{X^4}{M^4}\phi^5 - 4\frac{X^2}{M^2}(m+\varepsilon)\phi^3 - (2m+\varepsilon)\varepsilon\phi = 0 \tag{51}$$

so that at high densities it reduces to (40) with solution (43) but at lower densities it becomes

$$\nabla^2 \phi - (2m+\varepsilon)\varepsilon\phi = 0 \tag{52}$$

which in the stationary spherical case has

$$\phi = \frac{\sin\left(r\sqrt{\varepsilon|\varepsilon+2m|}\right)}{r\sqrt{\varepsilon|\varepsilon+2m|}} \quad \text{and} \quad \phi = \frac{\cos\left(r\sqrt{\varepsilon|\varepsilon+2m|}\right)}{r\sqrt{\varepsilon|\varepsilon+2m|}} \tag{53}$$

as solutions. This is the solution of a second-order derivative equation for the module with no chiral angle and as such it does not strictly describe a complete solution for the spinor field. It does however provide valuable insight into the dynamics of matter distributions that behave as bouncing droplets. The droplet itself would be the high-density part and the surrounding bath would be the low-density part of the module of the material field.

Could this really mean that the behaviour of quantum particles, so accurately mimicked by a bouncing droplet, is ultimately the result of the presence of torsionally induced tensions within matter distributions?

Quantum mechanics has always had difficulties in interpreting what a particle actually is. From Louis de Broglie on, the idea of particles behaving as waves was ubiquitously accepted, although still today we have no idea how exactly such a particle-wave duality really describes what happens in nature. It was de Broglie in a number of articles during the 1920s who put forward the possible interpretation known as pilot-wave model, with physical waves propagating in space and then guiding all particles around. Later known also as the double-solution model due to the fact that in it waves and particles are solution of two different equations [23,24], this interpretation has also been re-discovered by Bohm [25], and it is nowadays one of the few models still compatible with all restrictions imposed by Bell-like inequalities, from the non-locality to the contextuality constraints.

The fact that in the dBB interpretation of quantum mechanics contextuality, and more specifically non-locality, can be compatible with relativistic invariance has been recently shown be presenting a fully covariant version (with spin) of the dBB model. The key issue was to convert the Dirac theory into a consistent hydrodynamic formulation, which can be done by employing the polar form of spinor fields [26].

In this hydrodynamic formulation, quantum mechanics is naturally written in a manner that makes it easy to visualize and interpret the motion of particles as the result of an underlying wave dynamics, precisely as mimicked by bouncing droplets.

In the dBB model, waves and particles are seen as solution of two fields equations, a linear equation which possesses wave solutions and a non-linear equation which has soliton solutions. However, one is left with the uncomfortable feeling that it is inelegant to have such a duplicity, and on top of it one may still wonder why one equation should be linear and the other should be non-linear. A possible escape from this conundrum might come from the simple observation that extended waves and localized solitons do not need to be solutions of two different field equations, but just two different solutions of the same field equation with the wave behaviour emerging for low densities and the solitonic behaviour emerging for high densities. In this way, the most elegant solution to the problem raised by de Broglie one century ago would merely be to consider spinorial matter field equations in presence of torsion. As also commented in the previous section, the torsion mass, providing a natural scale for the matter distribution, would give the scale beyond which the quantum field would start to convert its wave behaviour into a localized bulk of matter. Then, the size of the particle would give the cut-off scale presently used in all renormalization schemes of QFT.

Is this interpretation sensible? One good aspect is that all it seems to need is the presence of torsion, which is a natural ingredient of the most general Dirac spinorial field theory. On the other hand, more work must necessarily be done to see whether complete exact solutions to the non-linear field equation can actually be found.

We are confident that very soon we might include some preliminary assessment on this problem in a following paper.

## 6. Conclusions

In this paper, we have considered the polar formulation of the Dirac spinor field theory fully coupled to geometric general backgrounds to face the problem of singularities.

As a first step, we have discussed the effective approximation of torsion and the fact that within this approximation the torsion can be effectively substituted in terms of the spin axial-vector in all field equations. The result is torsionally-induced spin–spin interactions

of attractive character, exactly as we have in the Nambu–Jona-Lasinio model. We have discussed how, for the Hawking–Penrose theorem, this model provides the environment to violate the dominant energy condition, resulting in the avoidance of singularities for gravitational systems constituted by a large number of particles. We have compared our results with previous attempts and given a physical justification.

For single particles, where all gravitational fields can be neglected, such singularity avoidance has been discussed by using the field equations for the module of the matter distribution. We have seen that second-order differential equations of Hamilton-Jacobi type for the module, in the effective approximation of torsion, and for large densities of matter, become one of the Lane–Emden equations for which an exact solution exists, and it is found to be non-singular. General considerations in QFT related to some ultraviolet divergences have been discussed in terms of a cut-off scale that is derived from the torsion mass.

Finally, general comments about solitons and bouncing droplets have been given in terms of covariant conditions. The relevance of these results for quantum mechanics, like the de Broglie double-solution model or the Bohmian dynamics, has also been addressed.

We are certain of the fact that the foundations of physics as a whole could benefit from the effects that would arise if torsion were considered as a fundamental field.

**Funding:** This research received no external funding.

**Data Availability Statement:** This research has no data available.

**Conflicts of Interest:** The author declares no conflict of interest.

## Notes

[1]   Dark Energy is also a problem that has to be faced, but in this case the treatment of the zero-point energy and the phase-shift due to spontaneous symmetry breaking are more pertinent to the domain of high-energy particle physics, and as such Dark Energy is more of an issue that belongs to the interface of cosmology and the fundamental constituents of matter.

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
