# Peer review of "Torsionally-Induced Stability in Spinors"

_universe, doi:10.3390/universe9020073_

Round 1
Reviewer 1 Report
The spin-torsion theory was first suggested by Cartan and then developed by Sciama and Kibble to include Dirac particles in Einstein's general relativity , and thus known as ECSK or ESK theory. In reference [15], the author employed the generalised ESK theory to solve the singularity problem stated by Hawking-Penrose theorem. In this paper, the author reformulates the previous results (in reference [15]) by virtue of the conceptually more clearer and computationally more powerful polar form of spinors. This method can provide deep insight into the relationship between the spin of source and the torsion of spacetime, so is very important.
By the way, the author says "if Dark Matter is indeed a form of matter, there is not a single prediction left to confirm", but dark energy is more problematic than dark matter, so GR cannot skirt around it.
Author Response
A comment about the cosmological constant has been added (see the modification in RED)
Reviewer 2 Report
The problem of interpreting quantum mechanics is inextricably linked with our understanding of the foundations of the structure of nature. On the other hand, the problem of singularities is another one of the difficult features of quantum field theory, and of course the renormalization procedure does not solve this problem at its core. The author considers the polar formulation of the Dirac spinor field theory fully coupled to geometric general backgrounds to face the problem of singularities. Guided by the main idea of this article, the author considers the influence of axial-vector torsion on the dynamics of the Dirac spinor fields. The author has demonstrated that the torsional correction has effects that make the otherwise singular spinorial matter field distribution regular. It would be correct if the author paid more attention to why the axial-vector part of the torsion was taken into account. Also, it would be good to write more information about the connection between the obtained results and the hydrodynamic description of quantum mechanics and the de Broglie-Bohm theory.
Author Response
- A comment has been added about the axial-vector part of the torsion tensor, with references supporting such a statement.
- Another comment has also been added about the hydrodynamic description of QM, again with a supporting reference.
(both additions have been highlighted in BLUE)